# Minimizing the Anticodon-Recognized Loop of *Methanococcus jannaschii* Tyrosyl-tRNA Synthetase to Improve the Efficiency of Incorporating Noncanonical Amino Acids

**DOI:** 10.3390/biom13040610

**Published:** 2023-03-28

**Authors:** Zhiyang Hu, Jinming Liang, Taogeng Su, Di Zhang, Hao Li, Xiangdong Gao, Wenbin Yao, Xiaoda Song

**Affiliations:** Jiangsu Key Laboratory of Druggability of Biopharmaceuticals, Department of Biochemistry, School of Life Science and Technology, China Pharmaceutical University, Nanjing 210009, China

**Keywords:** genetic code expansion, giant virus, anticodon recognition, loop minimization, noncanonical amino acids

## Abstract

In the field of genetic code expansion (GCE), improvements in the efficiency of noncanonical amino acid (ncAA) incorporation have received continuous attention. By analyzing the reported gene sequences of giant virus species, we noticed some sequence differences at the tRNA binding interface. On the basis of the structural and activity differences between *Methanococcus jannaschii* Tyrosyl-tRNA Synthetase (MjTyrRS) and *mimivirus* Tyrosyl-tRNA Synthetase (MVTyrRS), we found that the size of the anticodon-recognized loop of MjTyrRS influences its suppression activity regarding triplet and specific quadruplet codons. Therefore, three MjTyrRS mutants with loop minimization were designed. The suppression of wild-type MjTyrRS loop-minimized mutants increased by 1.8–4.3-fold, and the MjTyrRS variants enhanced the activity of the incorporation of ncAAs by 15–150% through loop minimization. In addition, for specific quadruplet codons, the loop minimization of MjTyrRS also improves the suppression efficiency. These results suggest that loop minimization of MjTyrRS may provide a general strategy for the efficient synthesis of ncAAs-containing proteins.

## 1. Introduction

Genetic code expansion (GCE), which utilizes the suppression of a developed orthogonal aminoacyl tRNA synthetase (aaRS)/tRNA pair to a stop codon, enables the site-specific incorporation of ncAAs into the interested proteins [1,2]. Compared to chemical synthesis, genetic code expansion can facilitate the incorporation of ncAAs within living cells. The ability to site-specifically incorporate ncAAs in vivo, with unique chemical functionalities, has profoundly simplified the study of complex biological processes. GCE technology has been found to have several applications, including the regulation of protein function via ncAAs [3,4,5], biochemical process indication by ncAA probes [6], research on biophysical processes [7], post-translational modifications [8], protein visualizations [9], pharmaceutical modifications [10,11], and treatment strategies [12,13]. At present, GCE is not only used to incorporate a variety of ncAAs into specific proteins at the prokaryotic level but has also rapidly developed in eukaryotic cells and animals, such as *C. elegans* [14], *D. melanogaster* [15], and *M. musculus* [16]. GCE has been proven to be a powerful and convenient tool for studying multiple biological problems at various levels of biological systems.

Although ncAAs can be integrated into proteins with high fidelity via GCE, the expression level of proteins containing different ncAAs, which are incorporated by specific aaRS variants, varies greatly. The expression of proteins containing ncAAs mediated by some aaRSs is very low. This disadvantage limits the further application of GCE, making optimizing and improving system efficiency a research priority. Numerous approaches have been developed to enhance the expression of interested proteins [17,18,19]. Takatsugu et al. rationally designed the anticodon-recognition interface of MjTyrRS and mutated the key residue Asp286 to Arg, which exhibited eightfold greater activity than the wild-type [20]. Miriam et al. diversified five residues at the MjTyrRS-tRNA_CUA_ anticodon recognition interface and obtained two mutants through screening, which exhibited about 1.5-fold higher GFP expression than wild-type MjTyrRS [21]. In 2009, Travis et al. constructed the dual-aaRS promoter system pEVOL, containing both the constitutive promoter glnS’ and the inducible promoter pBAD, leading to a mutant protein with an expression level of more than 100 mg/L [22]. In addition, the competition between amber-suppressor tRNAs and release factors causes inefficient amber suppression. The deletion of RF1 from the genome was described in previous works [23,24,25,26,27,28,29]. Wang et al. created the orthogonal ribosome, ribo-X, in *E. coli*, which can also minimize the effect of RF1 and improve the decoding of amber codons by tRNA_CUA_ in orthogonal mRNA [30]. These approaches have significantly improved the efficiency of incorporating ncAAs.

In typical viruses, the genome does not contain genes associated with the translation machinery. However, in giant viruses, whose genomes are larger than 200 kb, partial aaRSs were found [31]. It is probable that the aaRSs carried by the giant viruses help them intercept the host translation system after infection and initiate the synthesis of their related proteins more quickly. Previous research on *mimivirus* revealed that the virus TyrRS could recognize the yeast tRNA [32]. The mutation of discriminator bases confirmed that it could not recognize the *E. coli* tRNA. Owing to the unique position in evolution, the aaRSs of giant viruses may have unique properties and potentials that differ from those of the three kingdoms. Therefore, we studied the aaRSs of giant viruses with the aim of discovering aaRSs with new properties and functions. We used a bioinformatic approach to mining aaRSs from giant viruses and compared them with those from other species for analysis. A bioinformatics approach to comparing homologous aminoacyl-tRNA synthetases was used for the discovery of aaRSs. For example, Atsushi et al. designed *Candidatus* methanomethylophilus alvus PylRS (CMaPylRS) variants to incorporate Nε-benzyloxycarbonyl-L-lysine (ZLys) derivatives into proteins based on the alignment of the sequences of homologous PylRSs from different species [33]. Additionally, the mutation residues of previously developed homologous PylRS variants are frequently employed as a guide when generating new *Methanosarcina mazei* PylRS (MmPylRS) or *Methanosarcina barkeri* (MbPylRS) variants [34,35].

In this work, we performed gene mining on giant viruses. By aligning the gene sequences of TyrRSs on the reported giant virus genomes, we discovered the structural deficiencies in the anticodon recognition loop in partial virus TyrRSs, which deactivated the stacking mode into an attachment mode. The activity of giant virus’ TyrRS and MjTyrRS was tested in vivo, and the smaller loop of MVTyrRS was found to be highly favorable for recognizing triplet and quadruplet codons and approximately three times more effective than MjTyrRS. We transplanted the minimized loop to MjTyrRS to create three mutants, which were 1.8–4.3-fold more efficient than the wild-type MjTyrRS. In addition, we applied it to a variety of MjTyrRS variants, resulting in a strengthened effect ranging from 15–150%. The loop minimization mutation also displayed some benefits to the quadruplet codons, and this effect was most pronounced in the C321.∆A.exp∆PBAD strain. After loop minimization was applied to the MjTyrRS variants, the suppression efficiency of quadruplet codons exhibited a similar enhancement effect to that of the amber codon.

This study optimized the ncAA incorporation of MjTyrRS by referring to the natural TyrRSs of giant viruses. It was found that, in general, reducing the anticodon recognition loop and tuning the recognition mode could increase the expression of proteins containing ncAAs. Our work not only effectively improves the incorporation efficiency of MjTyrRS/MjtRNA but also provides a general strategy for many developed MjTyrRS variants or other aaRS/tRNA pairs from archaea, whether their introduction activity is high or low to increase the affinity with tRNA_CUA_ and improve the yield.

## 2. Materials and Methods

### 2.1. Materials

The noncanonical amino acids 4-Borono-L-phenylalanine (CAS 76410-58-7), 4-Nitro-L-phenylalanine (CAS 949-99-5), and 4-Azido-L-phenylalanine (CAS 33173-53-4) were purchased from Aladdin, Shanghai, China. ^14^C-labeled tyrosine (cat.no. ARC0655-250uci) was purchased from American Radiolabeled Chemicals, Inc. The aminoacylation assay was completed in the Radioisotopic Laboratory of Nanjing Agricultural University.

Polymerase chain reactions (PCR) were performed using the 2 × Phanta Max Master Mix (Vazyme, Nanjing, China). The ClonExpress II One Step Cloning Kit was purchased from Vazyme, Nanjing, China. L-arabinose was purchased from Macklin. The Native PAGE Preparation Kit was purchased from BBI Life Sciences, Shanghai, China. The BugBuster^®^ Protein Extraction Reagent was purchased from Merck.

### 2.2. Bacterial Strains

The *E. coli* DH10B strain was purchased from Tsingke Biotechnology. The *E. coli* C321.∆A.exp∆PBAD strain [36] was previously constructed in our lab based on *E. coli* C321.∆A.exp.

### 2.3. Plasmids Construction

Plasmids pEvole-MjTyrRS(wt), pEvole-B(OH)_2_PheRS, pEvole-pCNPheRS1, and pBAD-eGFP150TAG-MjtRNA_CUA_ were obtained from the previous construction in our lab.

To construct the pEvole-MVTyrRS(wt), the sequence of MVTyrRS was obtained from the NCBI Genome Virus Database (https://www.ncbi.nlm.nih.gov/genome/viruses, accessed on 1 December 2021). The gene sequence was synthesized and constructed into the pEvole vector by Genewiz, Suzhou, China.

To construct pBAD-eGFP150(XXX)-MjtRNA(XXX), the site-directed mutagenesis primers (Appendix A) of the 150th codon of eGFP to other codons were designed. After digestion with DpnI, the PCR product was transformed into *E. coli* DH10B. Monoclonal colonies, grown overnight on LB plates, were picked to extract plasmids for sequencing. Then, the anticodon ring of MjtRNA was also mutated to the anticodon complementary to the 150th codon of eGFP using the above methods. Sequencing was performed to confirm the mutation.

To construct the loop minimization MjTyrRS plasmid, the reverse primers (Appendix A) for loop minimization mutagenesis were designed. The MjTyrRS plasmid that was to be modified was used as a template, and the corresponding loop minimization primers were added to perform the PCR amplification reaction. The resulting linearized MjTyrRS plasmid was then ligated using the ClonExpress II One-Step Cloning Kit.

### 2.4. Expression of Orthogonal eGFP in E. coli and Fluorescence Quantification

The *E. coli* DH10B cells containing the reporter plasmids pBAD-eGFP150(TAG)-MjtRNA and pEvole-MVTyrRS or pEvole-MjTyrRS were cultured overnight in LB medium supplemented with the required antibiotic at 37 °C and 220 rpm. The cells were grown until the OD_600nm_ was 0.8, and the expression of eGFP was induced with the addition of L-arabinose (0.2% final concentration) for 10 h. Then, the bacteria were harvested from an 800 µL suspension by centrifugation at 12,000× *g* rpm for 2 min at 4 °C. The cell pellet was resuspended in 200 µL BugBuster^®^ solution to lyse the cells. After 20 min, 600 µL of PBS was added, and 200 µL of this solution was used for fluorescence determination with an excitation wavelength of 480 nm and an emission wavelength of 530 nm. The fluorescence intensity was acquired with the Tecan Infinite 200PRO[AMS1] (Tecan, Grödig, Austria). The remaining bacterial solution was used to measure OD_600nm_. OD6_00nm_ was acquired with Molecular Devices SpectraMax 190 (San Jose, CA, USA).

### 2.5. Native PAGE Gel Electrophoresis

After overnight expression of eGFP, the bacteria were harvested from an 800 µL suspension by centrifugation at 12,000× *g* rpm for 2 min at 4 °C. Then, the supernatant medium was discarded. The cell pellet was resuspended in 800 µL ddH_2_O and centrifuged to discard the supernatant. This was repeated twice. The cell pellet was resuspended in 200 µL BugBuster^®^ solution to lyse the cells. After 20 min, 600 µL PBS was added to the solution. Then, 2x bromophenol blue loading buffer (1.25 mL Tris-HCl (0.5 M, pH 6.8), 3 mL glycerin, and 0.2 mL 0.5% bromophenol blue, 5.5 mL ddH_2_O) were added. Native PAGE gel was prepared with the Native PAGE Preparation Kit from BBI. The electrophoresis buffer (3.03 g Tris, 14.4 g Glycine, pH 8.8, fixed volume to 1 L) was added to the electrophoresis tank. An equal amount of sample solution was injected into the sample well using the syringe. Electrophoresis was performed at a constant voltage of 100 V and a low temperature. The gel was carefully peeled off and placed under a gel imager for fluorescence imaging. Native PAGE gels were imaged by Tanon 5200 Multi.

## 3. Results

### 3.1. Gene mining of Giant Viruses

To study the aaRSs from giant viruses, we conducted gene mining on all the reported giant virus genomes (Supporting Information datasheet). The sequences of 40 tyrosyl-tRNA synthetases were summarized and retrieved (Appendix A). Then, ClustalX was used to align the TyrRS gene sequences of giant viruses with those from other species. All of the TyrRS gene sequences of archaea were aligned. The results revealed that the TyrRSs from the giant viruses are highly homologous with those of archaea, but some of them have missing anticodon recognition loops.

As shown in Figure 1, the sequences of amino acids corresponding to the MjTyrRSs’ secondary structure are highly conserved. TyrRSs from giant viruses and MjTyrRS have similar and conservative secondary structural units. However, the linking sequences between the secondary structure units are less conserved. Their lengths vary widely, from several to dozens of amino acids (Appendix A). In addition, we found a difference at the tRNA binding interface corresponding to the η3 helix of MjTyrRS in the TyrRS sequences of some giant viruses. These sequences are lacking in the partial TyrRSs of giant viruses at the anticodon recognition interface.

According to phylogenetic analysis, this difference only appears at specific evolutionary branches, probably resulting from sequence deletion or mutation in specific species during long-term evolution (Figure 2). Because this region is located at the tRNA interaction interface, its amino acid types and space occupation have a significant impact on the affinity of the aaRS and tRNA. Similar sequence conservation and differences were not observed in the alignment with eukaryotic cells. We hypothesized that a closer genetic relationship exists between giant viruses and archaea.

### 3.2. Activity Assay of Mimivirus TyrRS in Response to Triplet Codons

To investigate the interesting sequence difference, we selected the MVTyrRS (PDB, 2J5B) from *Acanthamoeba polyphaga mimivirus* for further exploration. The sequence similarity between MVTyrRS and MjTyrRS was identified through the amino acid sequence alignment. We speculated that MVTyrRS might have a similar suppression activity to MjTyrRS. Therefore, we tested the amber suppression activity of MVTyrRS in vivo. The MVTyrRS gene was obtained and constructed into the plasmid vector pEvole. Then, we co-transformed the reporter plasmid pBAD-eGFP150(TAG)-MjtRNA_CUA_ with the pEvole-MjTyrRS(wt) and pEvole-MVTyrRS(wt) into the *E. coli* DH10B. The results showed that, in the DH10B strain, the fluorescence intensity of wild-type MVTyrRS was much higher than that of wild-type MjTyrRS by a factor of approximately three (Figure 3A).

Next, we tested the amber suppression activity of MVTyrRS to explore its performance in the RF1 knockout strain. The above plasmid transformations were repeated in *E. coli* C321.∆A.exp∆PBAD [36]. The results indicated that, in the C321.∆A.exp∆PBAD strain, the fluorescence intensity of the MVTyrRS was similar to that in the DH10B strain, as was the MjTyrRS (Figure 3C).

To determine the suppression efficiency of MVTyrRS in response to other nonsense codons, we mutated the 150th site of enhanced green fluorescent protein (eGFP) into different stop codons and the anticodon ring of MjtRNA_CUA_ into the corresponding complementary pairing anticodon. Then, the two modified genes were constructed into the plasmid pBAD-eGFP150-MjtRNA. The plasmids pBAD-eGFP150(XXX)-MjtRNA(XXX) were co-transformed with pEvole-MjTyrRS(wt) and pEvole-MVTyrRS(wt) into *E. coli* DH10B. The fluorescence intensity of eGFP was measured to compare the suppression efficiency of the aaRS in response to different codons. The results showed that, in DH10B, as in MjTyrRS, the suppression activity of MVTyrRS towards UGA was higher than that of UAA. However, the fluorescence intensity of MVTyrRS was higher than MjTyrRS. In the C321.A.expPBAD strain, the suppression results of MVTyrRS and MjTyrRS were similar to those in DH10B.

### 3.3. Activity Assay of Mimivirus TyrRS in Response to Quadruplet Codons

Considering the high levels of suppression in response to the termination triplet codons of MVTyrRS, the quadruplet mutation was performed at the 150th site of eGFP to detect the suppression activity of MVTyrRS in response to the quadruplet codons. This site was mutated into quadruplet codons starting with blank triplet codons, such as UAGN, UAAN, and UGAN. In addition, the CUA anticodon ring of MjtRNA was also mutated into the corresponding frame-shift suppressor tRNA that pairs with the above quadruplet codons. Then, the reporter plasmid was co-transformed with the aaRS plasmid into the *E. coli* DH10B and the C321.∆A.exp∆PBAD strains. The efficiency of MjTyrRS in recognizing quadruplet codons in the *E. coli* DH10B strain was determined. All quadruplet codons in the experimental groups exhibited low fluorescence intensity except for UAGA, which was slightly more intense than the others. The suppression efficiency measurements of MVTyrRS indicated that its responsiveness to the majority of quadruplet codons is comparable to that of MjTyrRS. UAGA is an exception to this rule, as its fluorescence intensity was nearly threefold that of MjTyrRS (Figure 3B).

Then, the assay of quadruplet codons recognition was conducted in the C321.∆A.exp∆PBAD strain. MjTyrRs exhibited significantly increased fluorescence towards several quadruplet codons, including UAGA, UAGU, and UAGG. However, the suppression efficacy of other quadruplet codons, such as UAAN and UGAN, did not significantly improve. MVTyrRS showed a high response activity to a subset of quadruplet codons, similar to MjTyrRS. Nevertheless, the suppression efficiency of MVTyrRS is significantly greater than that of MjTyrRS: 2.6 times that of UAGA, 3.3 times that of UAGU, and 7 times that of UAGG (Figure 3D). Although the MjTyrRS and the MVTyrRS evolved from different species, the responsiveness trends of the two for different codons are remarkably similar, and the suppression efficiencies of MVTyrRS in response to various codons are greater than those of MjTyrRS.

### 3.4. Crystal Structure Comparison of MVTyrRS

To investigate the structural causes of the high MVTyrRS activity, we conducted a structural analysis of MVTyrRS and MjTyrRS. The comparison of crystal structures of MjTyrRS (PDB, 1J1U) and MVTyrRS (PDB, 2J5B [32]) revealed the structural basis for the difference in efficiency (Figure 4A). In the crystal structure of MjTyrRS, base G34 of MjtRNA forms two hydrogen bonds with Asp286, which are conserved in archaea and eukaryotes. Residues Phe261 and His283 form a sandwich-like conformation with the base G34 in MjtRNA via a staking interaction. By aligning and comparing the crystal structures of MjTyrRS and MVTyrRS, we found that the crystal structure of MVTyrRS strongly resembles that of MjTyrRS. The four main structural regions of the two TyrRSs, the N-terminal regions, the Rossmann-fold domain, the CP1 domain, and the C-terminal domain, have a similar secondary structure arrangement. However, a significant difference is that MVTyrRS possesses a smaller loop structure instead of the larger one in which the Phe261 of MjTyrRS is located, which appears at the aaRS-tRNA binding interface. Enlightened by the unique structure of MVTyrRS, we conceived a novel approach referring to the MVTyrRS structure to minimize the MjTyrRS loop, which could optimize the interaction between MjTyrRS and MjtRNA and improve the efficiency of ncAA incorporation in response to the amber codon (Figure 4B). We constructed models of the MjTyrRS loop-minimized mutations using the homology modeling tool and compared them to the wild-type MjTyrRS structure. Due to the deletion of some amino acid residues, the minimized mutations exhibit a much smaller flexible loop structure. The interaction of tRNA with MjTyrRS changes from being inserted between Phe261 and His283 to being attached to His283, which may be more conducive to the recognition and binding of tRNA.

### 3.5. Activity Assay of Loop Minimized Wild-Type MjTyrRS

To confirm the feasibility of minimizing the loop for suppression promotion, we first minimized the MjTyrRS loop in its wild-type form. Referring to the sequence of MVTyrRS, 254-TIKRPEKFGGDL-265 of MjTyrRS was reduced to mutant Loop-1 (254-TLCGL-265), flexible strand Loop-2 (254-TGSGL-265), and Loop-3 (254-TGGSGL-265). The loop-minimized MjTyrRS plasmid was co-transformed with pBAD-eGFP(150TAG)-MjtRNA_CUA_ into *E. coli* DH10B. The fluorescence intensity of eGFP was measured to characterize the degree of amber suppression efficiency improvement obtained by reducing the loop. The results indicated that the fluorescence intensity of eGFP was greatly enhanced after loop minimization. Mutant Loop-1 showed the most noticeable improvement effect—4.4-fold more than wild-type MjTyrRS—approaching the fluorescence intensity of wild-type MVTyrRS. In addition, mutants Loop-2 and Loop-3 were improved to varying degrees, with Loop-2 increasing by 93% and Loop-3 increasing by 76% (Figure 5A). Furthermore, the native gels were consistent with the measured fluorescence value. We also performed the aminoacylation assays in vitro. The k_m_ values measured indicated that the K_m_ of the wild-type MjTyrRS in response to MjtRNA was approximately 2.5 times higher than that of the MjTyrRS Loop-1 mutant (Appendix A and Appendix A). This suggests that the loop minimization does contribute to the increased affinity between MjTyrRS and MjtRNA.

### 3.6. Activity Assay of Loop Minimized MjTyrRS Variants

After a significant increase was achieved in the wild-type MjTyrRS, loop minimization was implemented on the evolved MjTyrRS mutants. First, we reduced the loop of B(OH)_2_PheRS [37] that recognizes 4-B(OH)_2_-Phe with low efficiency. Three mutants showed apparent improvements in the fluorescence intensity after loop minimization modifications, with Loop-1 showing a maximum increase of 147%, Loop-2 of 134%, and Loop-3 of 59% (Figure 5B). The same mutation was then performed on MjTyrRS mutants with greater incorporation efficiency, and its effect on the incorporation activity of various ncAAs was evaluated. pCNPheRS [38] was modified and used to incorporate 4-Azido-Phe (Figure 5C) and 4- Nitro-Phe (Figure 5D). The results showed that only mutant Loop-1 showed an increase in fluorescence intensity—a 15% increase for the incorporation of 4-Azido-Phe and a 23% increase for the incorporation of 4-Nitro-Phe. Loop-2 and Loop-3 did not demonstrate a significant improvement in efficiency. The results for the native PAGE gel correspond to those for relative fluorescence intensity. The above results show that the amber suppression efficiency correlates with the spatial size of the anticodon recognition loop and can be improved by adjusting the loop space occupation. When used to engineer MjTyrRS mutants, this strategy has diverse effects on different mutants (Appendix A).

### 3.7. Quadruplet Codons Suppression of Loop Minimization MjTyrRS Variants

After demonstrating the potential of loop minimization to increase the affinity of the binding interface, we tested whether loop mutants can improve the efficiency of quadruplet codons, whose extensive application was previously limited. The MjTyrRS loop-minimized mutants were co-transformed with the pBAD-eGFP150(XXXX)-MjtRNA(XXXX) mentioned above into *E. coli*. In the DH10B strain, the fluorescence intensity of specific quadruplet codons, decoded by loop-minimized MjTyrRS, is slightly higher than that of wild-type MjTyrRS. The quadruplet codon UAGA, for example, is the most effective, with the Loop-1 mutant showing a 45% increase in suppression activity, Loop-2 a 37% increase, and Loop-3 a 132% increase compared to wild-type MjTyrRS (Figure 6A). However, the other quadruplet codons, except UAGA, did not show significant efficiency improvements after loop minimization. When the fluorescence assay was performed in the C321.A.expPBAD strain, as previously demonstrated by wild-type MjTyrRS, an obvious increase in fluorescence value was observed in response to the quadruplet codon UAGA (Figure 6C), compared to the fluorescence intensity of wild-type MjTyrRS in C321.∆A.exp∆PBAD, Loop-1, Loop-2, and Loop-3, which increased by 192%, 202%, and 181%, respectively. In addition, the loop-minimized mutants also showed varying degrees of improvements in suppression for some of the other quadruplet codons, such as UAGG and UAGU (Appendix A).

Considering the best suppression efficiency of wild-type MjTyrRS in response to the UAGA quadruplet codon, we co-transformed the pBAD-eGFP (150TAGA)-MjtRNA_UCUA_ with the MjTyrRS variant plasmid pEvole-pCNPheRS1 into the *E. coli* DH10B and C321.∆A.exp∆PBAD strains, respectively, to determine the efficiency of the loop minimization mutation for the incorporation of ncAAs into the protein by MjTyrRS in response to quadruplet codons. The experimental results showed that the read-through efficiency of the quadruplet codon in DH10B was greatly affected by the presence of RF1, and the expression of the full-length eGFP containing ncAA was low (Figure 6B). The loop minimization mutants helped to improve the incorporation of ncAAs, but fluorescence intensity remained at a relatively low level. In the C321.∆A.exp∆PBAD strain, in which RF1 is deleted, the expression of eGFP (150TAGA) was significantly improved (Figure 6D). The expression of eGFP (150TAGA) mediated by the pCNPheRS1 variant was slightly lower than that of eGFP (150TAG). The loop-minimized mutant Loop-1 showed some increment in the suppression efficiency of the UAGA quadruplet codon, with the incorporation of 4-Azido-Phe increasing by 15% and 4-Nitro-Phe increasing by 35%. However, mutant Loop-2 and Loop-3 did not increase the expression of full-length eGFP (150TAGA). Analogous to the previously observed effect on amber read-through efficiency, Loop-1 exhibited an improvement in MjTyrRS variants in response to the UAGA quadruplet codon.

## 4. Discussion

In this work, we proposed an approach to minimize the loop at the tRNA interaction interface of MjTyrRS to improve the affinity between MjTyrRS and MjtRNA, thereby increasing the efficiency of ncAA incorporation and the expression of desired proteins.

We performed the sequence alignment and phylogenetic analysis of TyrRSs in the reported genomes of giant viruses. At present, the discovered giant virus species are not well classified, and there is a multi-branch distribution in their evolution. One of these branches, represented by *Acanthamoeba polyphaga mimivirus*, is close to archaea in its genetic relationships. The giant viruses may originate from an ancient viral common ancestor that predated or coexisted with the last universal common ancestor (LUCA) [39,40] or from smaller viruses [41]. The aaRSs carried by giant viruses may derive from the aaRSs of ancient ancestors or may have been obtained from the host during evolution. This class of aaRS has been reported to recognize the eukaryotic yeast tRNA with up to 90% tyrosylation in previous reports [32]. The suppression assay in this work revealed that MVTyrRS could recognize the MjTyrRS orthogonal MjtRNA, and the efficiency of MVTyrRS is higher than that of wild-type MjTyrRS. By aligning the sequences, we found that the loop related to the anticodon recognition was missing, and we speculated that the loop was deleted during the natural evolution of giant viruses, possibly to increase the efficiency of the evolution process.

The suppression assays of MVTyrRS showed that MjtRNA that is orthogonal to MjTyrRS could be recognized. The suppression assays of MVTyrRS proved that MVTyrRS exhibits approximately three times the amber suppression efficiency of MjTyrRS, suggesting that the smaller loop contributes to increased activity. As with MjTyrRS, the suppression of the triplet stop codon by MVTyrRS was the highest for UAG, followed by UGA, and the lowest for UAA. However, the suppression efficiency of MVTyrRS for these stop codons was higher than that of MjTyrRS. In the C321.∆A.exp∆PBAD strain, the fluorescence intensity was not significantly increased compared to that in DH10B, possibly because the suppressor tRNA was efficiently expressed, making it hard for RF1 to compete for the amber codon, as has been reported before [42]. In the DH10B strain, MjTyrRS showed almost uniformly low activity for the quadruplet codons, except for a slightly higher activity for the UAGA, as did MVTyrRS. In the C321.∆A.exp∆PBAD strain, the quadruplet codons beginning with the amber codon showed a consistent increase in activity. As RF1 is knocked out in the C321.∆A.exp∆PBAD, competitive binding to the amber codon between RF1 and the suppressor tRNA is reduced, allowing for the translation machinery to express the full-length reporter protein. The suppression efficiency of the other quadruplet codons in the C321.∆A.exp∆PBAD was not boosted compared to that of DH10B. Since the termination of UGA and UAA is not affected by the deletion of RF1, these quadruplet codons may be bound and terminated by RF2 in the translation machine. Among the UAGN quadruplet codons, UAGA has the strongest suppression efficiency, UAGG, and UAGT are slightly less efficient, and UAGC is the weakest. The suppression efficiency of the quadruplet codons is consistent with previous reports [42]. The low efficiency of some quadruplet codons may be related to the weakened charging efficiency caused by anticodon mutations.

We compared the crystal structures of MjTyrRS and MVTyrRS and investigated the structural basis for the high activity of MVTyrRS. When the loop of the tRNA recognition region exists, as shown in the crystal structure of MjTyrRS, the G34 base of MjtRNA is inserted between Phe261 and His283 [20,43]. However, MVTyrRS deletes this structure, and the G34 is attached to His283. The loop at the recognition interface is perhaps not necessary for the binding of MjtRNA and MjTyrRS but rather a hindrance. The suppression activity assay in vivo showed that the efficiency of MVTyrRS was three times higher than that of MjTyrRS due to the absence of the loop. MVTyrRS, which is structurally similar to MjTyrRS, could recognize MjtRNA, suggesting that changes in interaction mode do not affect recognition between the aaRS and tRNA. A smaller loop structure could improve the suppression efficiency. Therefore, the deletion of the aaRS-tRNA interaction loop inspired us to develop a new strategy to improve the suppression activity of MjTyrRS, that is, to minimize the recognition loop of MjTyrRS to improve the affinity between MjtRNA and MjTyrRS. During the interaction between MjTyrRS and MjtRNA, the anticodon region of MjtRNA_GUA_ is a major recognition element of MjTyrRS. A G34C change in the tRNA would render the interaction between MjTyrRS and the resulting MjtRNA_CUA_ less optimal, which could lead to a decrease in recognition efficiency and the failure of efficient aminoacylation. In previous work, the single mutation D286R at the anticodon recognition interface of MjTyrRS improved the aminoacylation rate and the amber suppression efficiency [20]. Furthermore, at the MjtRNA interface, fine-tuning and the combination of many residue mutations were also implemented to improve amber suppression efficiency [44]. However, variations in the size of the entire loop structure have not been adopted to date. We minimized the anticodon recognition loop, switched the interaction mode from insertion to attachment, and found this improved the affinity between the MjTyrRS and the MjtRNA(Appendix A), providing a general strategy to improve efficiency.

Three loop minimization mutants were designed and applied to MjTyrRS. When the interaction loop of wild-type MjTyrRS was minimized, the efficiency of charging the canonical substrate was dramatically increased, demonstrating an effective improvement in affinity (Appendix A and Appendix A). In addition, we found that loop minimization causes a significant increase in the expression of MjTyrRS (Appendix A). When the developed MjTyrRS variants’ loop was minimized, loop minimization could also boost the efficiency of charging ncAAs to the specific site of a protein. However, the enhancement effect of developed MjTyrRS variants using this strategy is not well-established. Some loop minimization mutants, such as Loop-1, showed good improvements in the incorporation of ncAAs, while others did not. As indicated in the previous works, the MjTyrRS–anticodon interaction and the MjTyrRS–ncAA interaction mutually affect each other [44]. We speculated that the amino acid residue types in the loop affect the recognition efficiency when incorporating different ncAAs. Therefore, the size of the loop and the type of residues on the loop need to be comprehensively considered in subsequent studies. The investigation into the loop minimization strategy for MjTyrRS variants also revealed that certain variants exhibit a noteworthy increase in the background expression of desired proteins in the absence of corresponding ncAAs (Appendix A), which is detrimental to the application of GCE. This phenomenon may be attributed to the enhanced affinity between MjTyrRS and MjtRNA, which could lead to an increase in its background expression. This observation suggests that caution must be exercised when implementing this strategy, particularly with MjTyrRS variants exhibiting poor orthogonality. We also observed that when CNPheRS was subjected to loop minimization, its background expression was slightly reduced compared to the wild-type CNPheRS, which is in contrast to the other MjTyrRS variants whose background expression was usually elevated. We speculated that the impact of loop minimization on the background expression of CNPheRS might be unique and distinct from that of the other MjTyrRS variants, and it may be an isolated case.

The results of the loop minimization strategy applied to optimize the efficiency of wild-type MjTyrRS in response to quadruplet codons showed the expected improvement. Either Loop-1, which mimics the MVTyrRS sequence, or the flexible strands Loop-2 and Loop-3 can effectively improve the suppression efficiency in response to the specific quadruplet codons, mainly UAGA. When the loops of MjTyrRS variants were minimized, their improvement in the read-through of eGFP (150TAGA) to express the full-length protein containing ncAAs was similar to that of the amber codon. However, not all quadruplet codons have an efficiency-enhancing effect when decoded by a loop-minimized mutant. Except for UAGA, the suppression efficiency of UAAN, UGAN, and UAGN was generally low. As these quadruplet codons are intrinsically affected by release factors, it is hard to read through stop codons expressing full-length proteins in strains whose release factor has not been knocked out. The charging efficiency of tRNA will be affected by anticodon mutagenesis, impacting their suppression efficiency. While loop-minimized variants have different effects on the quadruplet codons than on the amber codon, the smaller space occupation on the interface is likely to increase the expression of the MjTyrRS (Appendix A). We determined the k_m_ values of quadruplet MjtRNA_UCUA_ and MjTyrRS and found that loop minimization did not significantly alter the affinity between the two (Appendix A). As the quadruplet MjtRNA_UCUA_ is not a natural substrate for MjTyrRS, and its interaction mode with MjTyrRS differs from that of MjtRNA_CUA_, we speculate that loop minimization increases the efficiency of the reaction due to the increased expression of MjTyrRS rather than the improved affinity between MjTyrRS and MjtRNA_UCUA_. Furthermore, the sequence of the reduced loop has different effects on different tRNA anticodon rings. At the tRNA binding interface, the mechanism of aaRS-tRNA-ncAA interactions plays an important role. If the contact interface is modified to improve suppression efficiency, the loop-minimized sequences corresponding to different anticodon rings need to be considered. In addition, the modifications of tRNA should be included in the scope of optimization.

## 5. Conclusions

By searching the TyrRS gene sequences from giant viruses and aligning them with that of other species, we found that the TyrRSs derived from giant viruses have high homology with archaeal MjTyrRS. By researching giant virus-derived MVTyrRS, a difference was confirmed based on the structure of the amber suppression efficiency between MjTyrRS and MVTyrRS. The loop minimization was conducted with reference to the structure of MVTyrRS and displayed an excellent effect, improving the incorporation efficiency of ncAAs. In addition, the loop minimization modification showed a great improvement in response to quadruplet codon suppression. Therefore, the loop minimization modification caused by giant virus sequence differences could provide a new strategy for optimizing the MjTyrRS/MjtRNA system, improving ncAA incorporation and protein expression.

## Figures and Tables

**Figure 1 biomolecules-13-00610-f001:**
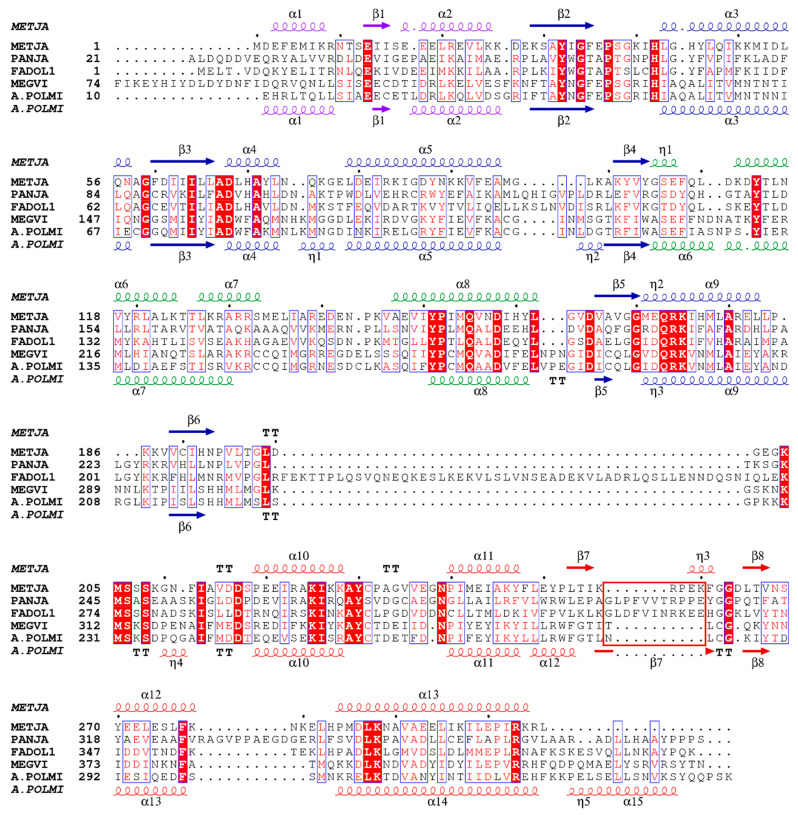
Sequence alignment of partial TyrRSs. The sequences are derived from *Methanococcus jannaschii*, *Pandoravirus japonicus*, *Fadolivirus 1*, *Megavirus vitis*, and *Acanthamoeba polyphaga mimivirus*. The secondary structural elements of the *M. jannaschii* and *A. polyphaga mimiviruses* are shown above and below the alignment as coils for α-helices and arrows for β-strands, respectively. The N-terminal regions, the Rossmann-fold domain, the CP1 domain, and the C-terminal domain are shown in violet, blue, green, and red, respectively. The residues conserved among the *M. jannaschii* and giant viruses are marked in red, and slightly fewer conservative residues are in red font and framed by blue boxes. Alignment was completed with ClustalX2. The figure was drawn with ESPript.

**Figure 2 biomolecules-13-00610-f002:**
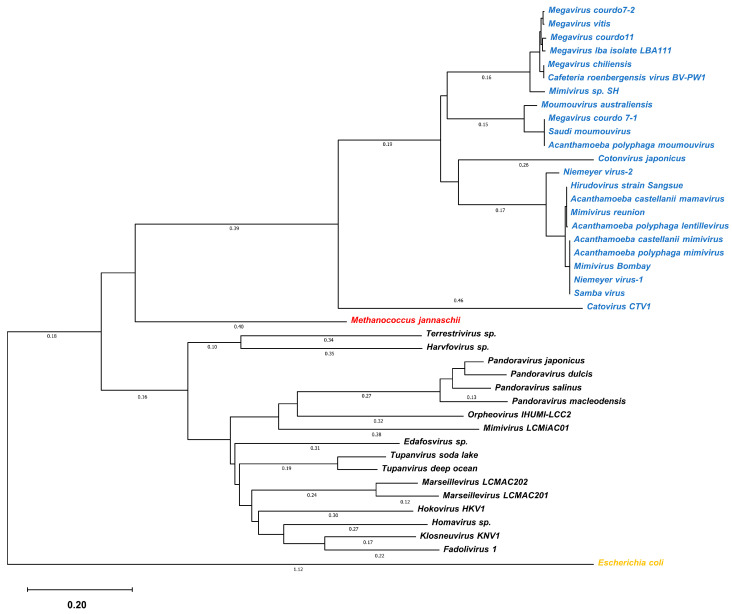
Phylogenetic analysis of 40 tyrosyl-tRNA synthetase genes identified from giant viruses. The names of the species whose TyrRSs lack the loop sequence are marked in blue. The other giant viruses, *Methanococcus jannaschii* and *E. coli* are represented in black, red, and purple, respectively.

**Figure 3 biomolecules-13-00610-f003:**
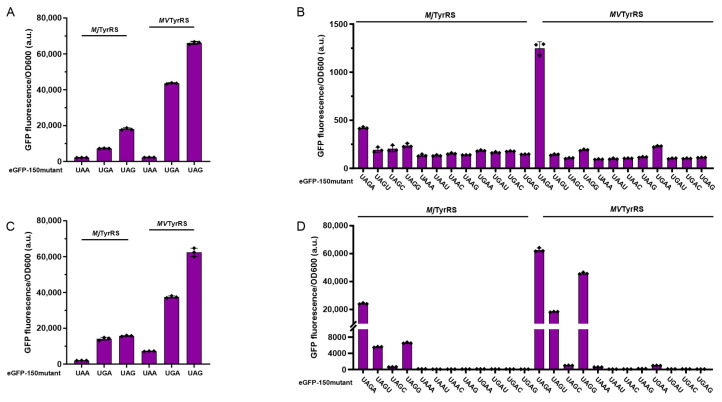
The suppression activity of nonsense and quadruplet codons by MVTyrRS and MjTyrRS. (**A**) Analysis of amber suppression activity of the MVTyrRS and MjTyrRS for different nonsense codons in DH10B strains by eGFP reporter assay. (**B**) Quadruplet codons in DH10B strains. (**C**) Analysis of amber suppression activity of the MVTyrRS and MjTyrRS for different nonsense codons in C321.∆A.exp∆PBAD strains by eGFP reporter assay. (**D**) Different quadruplet codons in C321.∆A.exp∆PBAD strains. Error bars represent ± standard error of the mean from three biologically independent experiments.

**Figure 4 biomolecules-13-00610-f004:**
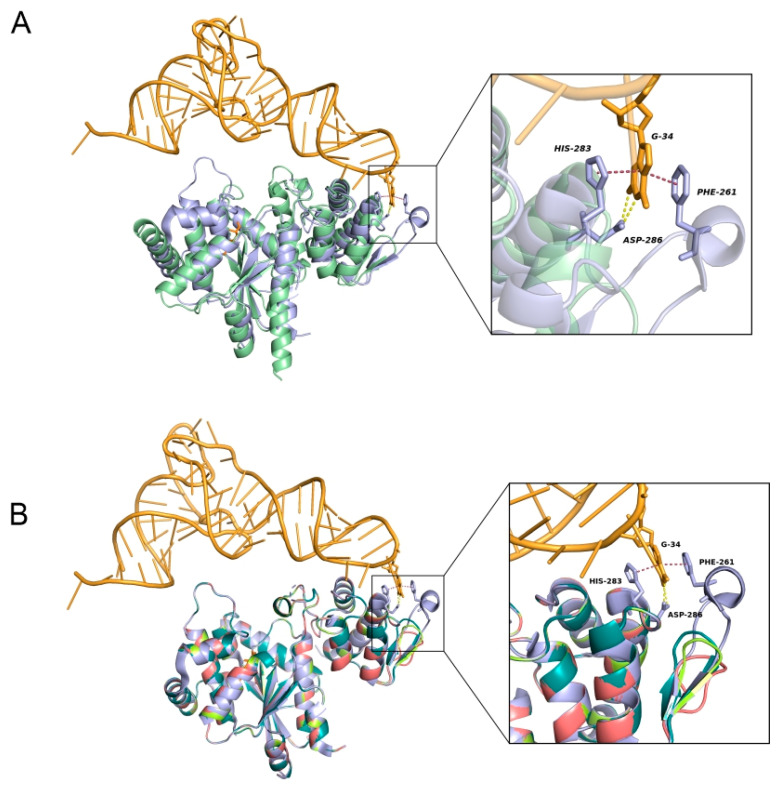
Crystal structure comparison. (**A**) Comparison of protein crystal structures between MVTyrRS and MjTyrRS. MjTyrRS is marked in light purple and MVTyrRS in light green. (**B**) Diagram of loop reduction modification of MjTyrRS, wild-type MjTyrRS, mutant Loop-1, Loop-2, and Loop-3 are shown in light purple, rose, light green, and dark green, respectively (structural simulation by SWISS-MODEL).

**Figure 5 biomolecules-13-00610-f005:**
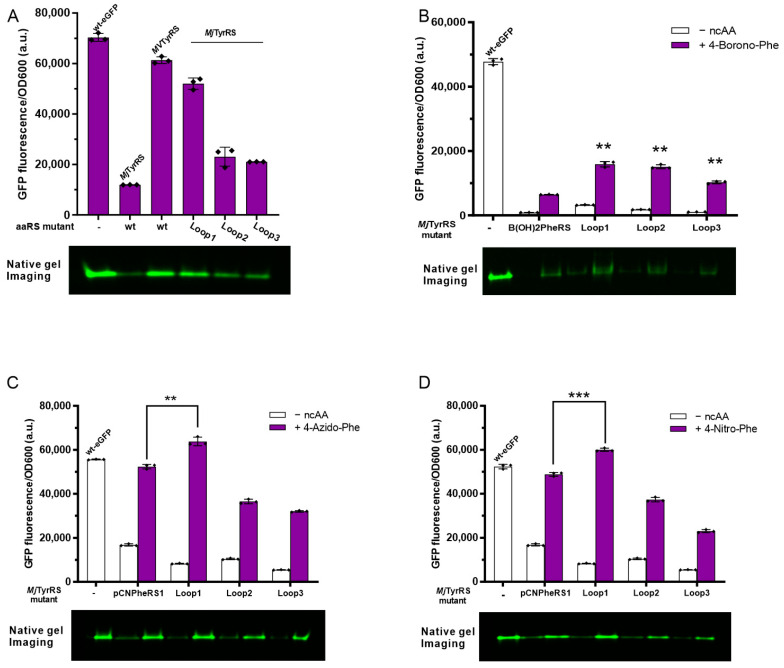
The loop minimization of MjTyrRS. (**A**) Analysis of amber suppression activity of the different wild-type MjTyrRS loop-minimized mutants by eGFP reporter assay and a native in-gel fluorescence imaging assay. eGFP-wt and MVTyrRS-wt are used as controls. (**B**) Comparison of fluorescence intensities of pB(OH)_2_PheRS loop-minimized mutants to incorporate 4-Borono-Phe into eGFP. (**C**) Comparison of fluorescence intensities of pCNPheRS1 loop-minimized mutants to incorporate 4-Azido-Phe. or (**D**) 4-Nitro-Phe into eGFP. Error bars represent ± standard error of the mean from three biologically independent experiments. Statistical significance is quantified with ordinary one-way ANOVA (** *p* < 0.01, *** *p* < 0.001).

**Figure 6 biomolecules-13-00610-f006:**
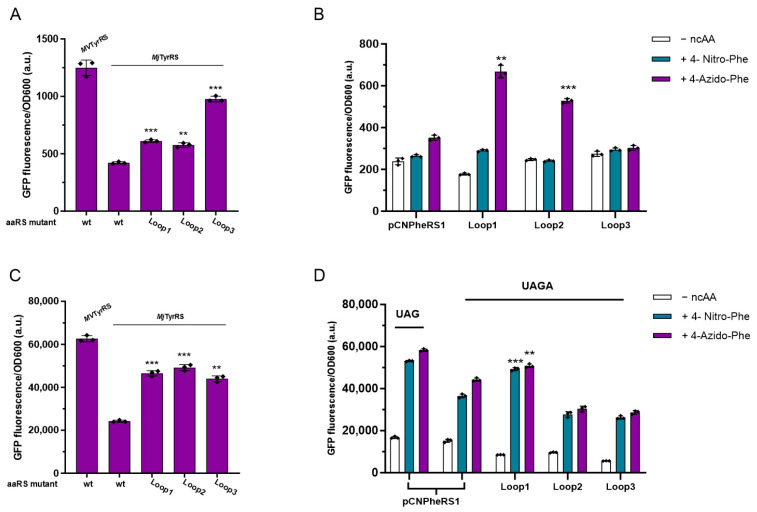
The suppression assay of loop minimization variants in response to the UAGA quadruplet codon In the *E. coli* DH10B strain, the suppression assay of (**A**) wild-type MjTyrRS loop minimization mutants and (**B**) MjTyrRS pCNPheRS1 loop minimization variants in response to UAGA quadruplet codon. In the *E. coli* C321.∆A.exp∆PBAD strain, the suppression assay of (**C**) wild-type MjTyrRS loop minimization mutants and (**D**) MjTyrRS pCNPheRS1 loop minimization mutants in response to UAGA quadruplet codon. Error bars represent ± standard error of the mean from three biologically independent experiments. Statistical significance is quantified with ordinary one-way ANOVA (** *p* < 0.01, *** *p* < 0.001).

## Data Availability

Data are contained within the article and are available on request from the corresponding author, xiaoda.song@cpu.edu.cn (X.S.). Bacterial and Plasmid could be obtained from addgene. (pEvole-Mj-TyrRS(wt)-Loop1, addgene No. 197898; pBAD-eGFP150TAG-MjtRNA, addgene No. 197897; C321.∆A.exp∆PBAD, addgene No. 197896; pEvole-MV-TyrRS, addgene No. 197895.).

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
