# Peer review of "Minimizing the Anticodon-Recognized Loop of Methanococcus jannaschii Tyrosyl-tRNA Synthetase to Improve the Efficiency of Incorporating Noncanonical Amino Acids"

_biomolecules, 2023, doi:10.3390/biom13040610_

Round 1

Reviewer 1 Report

In this manuscript, the authors reported the minimization of the anticodon-recognition loop in the M. jannaschii tyrosyl–tRNA synthetase, used for improving the incorporation efficiency of noncanonical amino acids. The authors found that one of the MjTyrRS mutants, inspired by MVTyrRS, identified from giant viruses, improved the incorporation efficiency of noncanonical amino acids through amber suppression and UAGA quadruplet suppression. The authors suggested that the mutation would improve the affinity between TyrRS and the suppressor tRNA; thus, it can generally apply various MjTyrRS mutants that have been developed for various kinds of noncanonical amino acids.

The authors wrote that improving the affinity between TyrRS and the suppressor tRNA would cause improved incorporation efficiency, although the Km values were never determined in this study. The Km values of MVTyrRS against various tRNAs were determined by other researchers (ref. 32); however, this data cannot be applied to the MjTyrRS loop 1 mutant. Therefore, it is important to determine the Km value of the loop 1 mutant of the wildtype MjTyrRS as well as CNPheRS and B(OH)2PheRS against jannaschii tRNATyr and the amber/ UAGA quadruplet suppressor tRNAs.

Other comments for improving the manuscript are:

(1)  Are the expression level and the solubility of the loop 1-3 mutants similar to the wildtype MjTyrRS? The increased concentration of the functional MjTyrRS can improve the incorporation efficiency of noncanonical amino acids. If possible, analyze the soluble and insoluble fractions of the MjTyrRS mutants expressed in E. coli using SDS-PAGE.

(2)  Page 8 line 263: add ref 32 for PDB 2J5B.

(3)  Figure 3A and 3C are not mentioned in the main text.

(4)  In Figure 5D, why did the CNPheRS loop 1 mutant show lower GFP fluorescence intensity than the wildtype CNPheRS without ncAA? Please comment on this in the discussion.

Reviewer 2 Report

This paper by Song and colleagues aimed to improve the activity of the tyrosyl-tRNA synthetase from Methanocaldococcus jannaschii (MjTyrRS). MjTyrRS is an aminoacyl-tRNA synthetase that is commonly used to install unnatural amino acids into proteins in living cells. However, MjTyrRS interacts with the anticodon of its tRNA, which reduces aminoacylation efficiency when the anticodon of the tRNA is mutated. To address this limitation, the authors compared the sequence and structure of MjTyrRS to closely related TyrRSs from giant viruses and found that the giant virus enzymes lack the anticodon recognition motif. They speculated that by removing or shortening the MjTyrRS anticodon recognition motif, they could create mutants with improved activity. Indeed, by shortening the motif, the authors generated MjTyrRS variants that have altered activity and specificity towards tRNAs with diverse anticodons.

While the study is compelling and may be of interest to a broad audience involved in genetic code expansion and tRNA/aminoacyl-tRNA synthetase engineering, there a number of critical issues that must be addressed.

1.     The manuscript contains several confusing or difficult-to-read portions due to incorrect grammar or language usage.

2.     There are formatting issues that make the manuscript difficult to read, such as using abbreviations without defining terms (e.g., MmPylRS, CMaPylRS, DhPylRS, ZLys) and using inconsistent abbreviations and nomenclature (e.g., "MmPylRS" and "mmPylRS" are used interchangeably). Additionally, a single species is referred to as both Methanocaldococcus jannaschii and Methanococcus jannaschii, which is confusing to readers.

3.      "changing the recognition mode from insertion to attachment,"; it is not clear what this phrase is referring to.

4.     In Section 3.2, experimental results obtained with wild-type enzymes are discussed, however, the reader is not directed to the relevant data which is presumably shown in Fig. 3A?

5.     Increases in activity are reported with three significant digits (e.g., 2.58 and 3.29), but this level of precision is likely inaccurate. 

6.     Some MjTyrRS mutants developed herein show a significant increases in the yield of protein in the absence of the desired unnatural amino acid. This "background" is often considered detrimental in genetic code expansion applications. These results are buried in the supplementary information, but should be (at minimum) discussed in the main text.

7.     Controls without unnatural amino acids should be included for the four-base codons (at least in the supplementary information). This will clarify whether the increase in four-base codon suppression is due to the unnatural amino acid or whether the codon is simply better recognized by an endogenous tRNA.

8.     It is claimed that mutating MjTyrRS increases its affinity for the tRNA, but no measurements of affinity were made. The only assays performed were stop codon readthrough assays using a single reporter (eGFP). Any claims about the affinity of the tRNA and enzyme are speculative. While such speculations and hypotheses are appropriate, it should be made clear that they are such.

9.     Similarly, the authors refer to an “aminoacylation assay”. Aminoacylation assays must be done with purified enzymes and tRNAs. Again, the only assay that was done in this study was a stop codon (or four base codon) readthrough using a single reporter (eGFP). Any reference to aminoacyltion is only speculation since no aminoacylation assays were done. Speculations and hypothesis should be clearly described as such.   

Round 2

Reviewer 1 Report

Comment 1: The authors wrote that improving the affinity between TyrRS and the suppressor tRNA would cause improved incorporation efficiency, although the Km values were never determined in this study. The Km values of MVTyrRS against various tRNAs were determined by other researchers (ref. 32); however, this data cannot be applied to the MjTyrRS loop 1 mutant. Therefore, it is important to determine the Km value of the loop 1 mutant of the wildtype MjTyrRS as well as CNPheRS and B(OH)2PheRS against jannaschii tRNATyr and the amber/ UAGA quadruplet suppressor tRNAs.

Reply: Thank you for your valuable suggestion. We constructed the plasmids pET28a-MjTyrRS-6His and Pet28a-MjTyrRS(Loop1)-6His. Then we expressed and purified wild-type MjTyrRS and its loop-minimized mutants, as well as obtained MjtRNACUA by in vitro transcription. We determined the Km values of wild-type MjTyrRS and the MjTyrRS Loop-1 mutant against MjtRNACUA. The specific experiment methods can be found in the supporting information. The results showed that the Km value of the Loop-1 mutant was significantly lower compared to the wild-type MjTyrRS. The Km value reflects the affinity between MjTyrRS and MjtRNA, and the km of wild-type MjTyrRS is about 2.5-fold higher than that of the MjTyrRS Loop-1 mutant.

However, due to the unavailability of 14C-labeled ncAAs, we have not determined the Km values of the Loop-1 mutant against MjtRNAUCUA and CNPheRS and B(OH)2PheRS against MjtRNACUA. Based on the results of the aminoacylation assay we measured, we speculate that the affinity of the loop-minimized mutants for quadruplet codon suppressor tRNA and the affinity of CNPheRS and B(OH)2PheRS against the amber codon suppressor tRNA are also increased.

Re-comment1:

It is good to see the Km values of the Loop-1 mutant and WT (Fig. S9). What is the difference between WT-1 and WT-2, and between Loop1-1 and Loop 1-2? Please write it in the figure caption. Please also show the fitting curve for determining the Km values for the tRNA.

What is a substrate? Please write it in the figure caption (I think it is tRNACUA). Please also write the horizontal and vertical axis labels properly.

I understand that the authors could not determine the Km vales of CNPheRS and B(OH)2PheRS due to the unavailability of 14C-labeled ncAAs. But the authors could still determine the Km value of the Loop-1 mutant and WT for MjtRNAUCUA.

Comment 2: Are the expression level and the solubility of the loop 1-3 mutants similar to the wildtype MjTyrRS? The increased concentration of the functional MjTyrRS can improve the incorporation efficiency of noncanonical amino acids. If possible, analyze the soluble and insoluble fractions of the MjTyrRS mutants expressed in E. coli using SDS-PAGE.

Reply: We gratefully appreciate your comment. We have performed the SDS-PAGE of the MjTyrRS and loop-minimized mutants to analyze their soluble and insoluble fractions. We harvested the E. coli by centrifugation and lysed the cell pellet with the BugBuster® solution. The supernatant and precipitate were separated by centrifugation. Then we prepared samples of soluble and insoluble fractions for SDS-PAGE, respectively. The result of SDS-PAGE showed that, compared to wild-type MjTyrRS, there is no significant increase in the soluble fraction of loop-minimized mutants. On the basis of the pEvole-MjTyrRS, we constructed the plasmid whose MjTyrRS sequence was followed by the HA tag. Then we carried out Western Blot for further validation. The WB results showed that the protein expression of mutants was increased to a varying degree compared to that of the wild-type MjTyrRS. However, considering the amplification effect of WB and the previous results of SDS-PAGE, we believe that loop minimization has led to a minor improvement in the expression level of MjTyrRS.

Re-comment 2:

I do not understand that the comment “. However, considering the amplification effect of WB and the previous results of SDS-PAGE ••• we believe that loop minimization has led to a minor improvement in the expression level of MjTyrRS.”. The data in Figure S6 showed significant solubility improvement for the loop 1 and loop 2 mutants (lane 2 vs lanes 4 and 6). There is clear difference, so I cannot agree with the author's conclusion.

Reviewer 2 Report

The authors have addressed all my concerns. This is a nice paper that I can now endorse. 

Author Response

Thank you again for your comments and suggestions  for improving this manuscript.